# Bevacizumab as Single Agent in Children and Teenagers with Optic Pathway Glioma

**DOI:** 10.3390/cancers15041036

**Published:** 2023-02-07

**Authors:** Pierluigi Calò, Nicolas Pianton, Alexandre Basle, Alexandre Vasiljevic, Marc Barritault, Pierre Aurélien Beuriat, Cécile Faure-Conter, Pierre Leblond

**Affiliations:** 1Department of Pediatric Oncology, Institute of Pediatric Hematology and Oncology, 69008 Lyon, France; 2Department of Ophtalmology, Edouard-Herriot Hospital, 69003 Lyon, France; 3Department of Radiology, Leon Bérard Center, 69008 Lyon, France; 4Department of Medical Biology and Pathological Anatomy, Hospices Civils de Lyon (HCL), 69777 Bron, France; 5Department of Pediatric Neurosurgery, Hôpital Femme Mère Enfant-France (HCL), 69500 Bron, France

**Keywords:** bevacizumab, optic pathway glioma, pediatric low-grade glioma

## Abstract

**Simple Summary:**

Nowadays, there is no univocal therapeutical care for children with optic pathway gliomas (OPG): different chemotherapy regimens are proposed, but no one has clearly proved its superiority over the others on the PFS (Progression free survival). The efficacy of bevacizumab, an anti-VEGF monoclonal antibody, used in combination with Irinotecan, has been raised by several recent publications. However, Irinotecan has demonstrated side effects, especially digestive. Our main goal is to understand if bevacizumab could be efficacious used as a single agent against OPG.

**Abstract:**

This is a retrospective study conducted on patients with OPG, aged less than 19 years, treated with bevacizumab as a single agent, since 2010 at IHOPe (Institute of Pediatric Hematology and Oncology). Efficacy of the treatment was evaluated on the tumor response rate on MRI with a centralized review basing upon RAPNO criteria and with visual assessment basing upon a 0.2 log change in the logMAR scale. Thirty-one patients with OPG have been included. From a radiological point of view, best anytime responses were: 1 major response, 6 partial responses, 7 minor responses and 14 stable diseases; achieving disease control in 28 (96%) out of 29 patients. Ophthalmological response was evaluated in 25 patients and disease control was achieved in 22 (88%) out of 25, with 14 steady states and 8 significant improvements. Among patients treated with chemotherapy after the bevacizumab course, nine relapsed and have been retreated with objective responses. Bevacizumab used as single agent seems effective in children and adolescents with OPG. Our work paves the way for a phase II study in which bevacizumab alone could be used as frontline therapy.

## 1. Introduction

### 1.1. Pediatric Low-Grade Glioma and Optic Pathway Glioma (OPG)

Low-grade gliomas (LGG) are the most common central nervous system (CNS) tumor among children, accounting for approximately 30% of pediatric brain tumors [1]. They represent a very heterogeneous group of tumors and are defined by the World Health Organization as grade 1 or grade 2 tumors [2]. Among all LGG, optic pathway gliomas (OPG) affect, specifically, the pre-cortical visual pathway and they can occur either sporadically or in association with the tumor predisposition syndrome Neurofibromatosis type 1 (NF1) [3], with a prevalence of about 15% according to a longitudinal study [4]. OPG may involve the optic nerve in 25–35% of cases, or they may have a chiasmatic or post-chiasmatic localization. The localization on the optic nerve can be subdivided into orbital, intracanalicular and intracranial pre-chiasmal lesions [5]. Others tumors’ localizations occurs in the hypothalamus, or anterior third ventricle [6]. These different localizations have led to the modified Dodge classification [7], identifying anatomical sites, hypothalamic involvement, metastasis and NF1 status. Even if some patients experienced severe visual impairment, the clinical presentation is often characterized by a slow and indolent evolution, with a large proportion of patients being asymptomatic, particularly among patients with NF1.

When clinical symptoms are present, strabismus is the most commonly observed symptom, although patients may also present with visual loss, proptosis, nystagmus and diencephalic syndrome [8,9]. OPG should be evoked in any child presenting with unexplained visual loss, *spasmus nutans* type nystagmus, diencephalic syndrome or optic nerve atrophy. The ophthalmologic assessment is crucial and needs to be particularly adequate, since preservation of vision is a critical goal of the management of OPG. According to a retrospective study on 59 pediatric patients with sporadic OPG, a majority of pediatric patients had significant long-term visual impairment [10]. Along with visual assessment, imaging is critical in the diagnosis and management of OPG. The diagnosis is usually made following a clinical examination and a magnetic resonance imaging (MRI). On MRI, OPGs are usually hypo to iso-intense on T1, and hyperintense on T2 sequences [11]. Bright enhancement of the lesion is seen in more than 50% of tumors after gadolinium injection [9]. Performing a biopsy is unnecessary in the case of tumors with typical clinical characteristics and imaging findings in NF1 patients [9]. The molecular landscape of pediatric LGG involves somatic driver alterations that result in activation of the MAPK pathway and rearrangements involving *BRAF* gene [1]. Together, *KIAA1549-BRAF* transcript fusion, BRAFV600E mutation and *NF1* mutations account for 2/3 of pediatric LGG [12].

### 1.2. Management of OPG 

There is no consensus on the optimal management of childhood LGG: the decision to treat a patient with OPG depends on age, NF1 status, tumor size, tumor localization and, most significantly, on the impact of the tumor on neurological and visual functions and consequent functional deficits. The choice of treatment (wait and see, surgery, radiotherapy, chemotherapy or possibly targeted therapy) is one of the most challenging and controversial aspects of the disease, although current consensus is to treat children with evidence of visual or neurological deterioration [9]. Surgery has an essential place in LGG. For OPG, the place of the surgery depends on the extension of the lesion and the visual status. Complete resection is only feasible when the tumor is confined to the optic nerve and associated with homolateral complete blindness. However, partial debulking is often possible when in case of lateral extension or extension within the third ventricle especially in case of intracranial hypertension due to the tumor itself or obstructive hydrocephalus. Radiotherapy has a clear anti-tumor effect, but its use is limited by the risk of significant late effects such as vasculopathy such as Moyamoya [13], neurological, neurocognitive and endocrine complications and radiation-induced second tumors [9]. Chemotherapy is the favored first-line treatment. A variety of different drug regimens have shown efficacy, achieving 5-year progression free survival (PFS) depending on the regimen: weekly vinblastine with a PFS of 53.2% [14]; SIOP LGG 2004 (vincristine- carboplatine) with a PFS of 46% [15]; thioguanine, procarbazine, lomustine and vincristine regimen with a PFS of 52% ± 5%) [16]. However, although these different chemotherapy protocols most often allow satisfactory control of the disease, their impact on visual function recovery is more controversial. The effect of chemotherapies regimens on visual acuity (VA) is often modest. According to the SIOP LGG 2004 prospective cohort study, children with and without NF1 demonstrated the same rate of VA improvement, stabilization or worsening; however, children with sporadic OPG had a poorer VA outcome [17]. The SIOPE NF1 OPG workshop identified factors present at diagnosis associated with unfavorable visual outcomes in NF1 patients treated with the SIOP LGG 2004 regimen. An unfavorable outcome was associated with the presence of multiple visual signs and symptoms, abnormal visual behavior, new onset of visual symptoms and optic atrophy prior to treatment. Instead, squint, posterior visual pathway tumor involvement and bilateral pathway tumor involvement showed borderline significance [18]. Recently, targeted therapies of the MAPK pathway, such as MEK or BRAF inhibitors, have been used with promising results and represent an option for salvage treatment in both sporadic and NF1-associated OPG [19]. MEK inhibitors have been employed in the treatment of progressive and recurrent LGG in children, with a 2-year PFS of up to 69% [20]. 

### 1.3. Angiogenesis and Bevacizumab Rationale in OPG 

Among new therapies, bevacizumab is a humanized monoclonal antibody directed against vascular endothelial growth factor (VEGF) [21]. Brain tumors and among-all low grade gliomas have been shown to express high levels of VEGF [22]. The expected mechanisms of action of Bevacizumab are tumor size stabilization/reduction and vision sparing. Recently, the effect of anti-VEGF treatment on nerve protection and function has been reported: by normalizing the tumor vascularization, anti-VEGF treatment is able to alleviate nerve oedema and deliver oxygen more efficiently to the nerve, thus reducing nerve damage and improving its function [23]. The efficacy of bevacizumab used in combination with conventional chemotherapy with irinotecan on 10 pediatric patients with recurrent LGG was first published in 2009 [24]. Since the cited Packers’ et al. paper, other pediatric series coupling bevacizumab and irinotecan have shown disease control on pediatric LGG [25,26]. Irinotecan is a camptothecin derivative that inhibits topoisomerase I, which crosses the blood–brain barrier, and it has shown activity against human glioblastoma cells with multidrug resistance [27]. However, irinotecan is often associated with significant and adverse gastrointestinal events requiring dosage adjustments, discontinuation of treatment, and change in treatment intervals. Conversely, it has been suggested that treatment with bevacizumab as monotherapy could be effective with limited toxicity [28]. Based on these assumptions, we decided to analyze the effect of a bevacizumab as a single agent on children with OPG. 

## 2. Materials and Methods

This was an observational retrospective monocentric study of patients aged less than 19 years old and with low-grade OPG, which took place between January 2010 and December 2020 at the Pediatric Hematology and Oncology Institute (IHOPe, Lyon, France). The study was conducted with institutional ethics approval. The information was collected by a standardized and anonymized data collection sheet. In accordance with French regulations, the study protocol was approved by the Commission Nationale de l’Informatique et des Libertés (CNIL) based on the declaration of conformity MR-004 (No. R201-004-216). Patients and their caregivers were informed about the study and they did not raise any objection. Participants have been identified through the register of the neuro-oncology multidisciplinary meeting of the Auvergne-Rhône Alpes region (RCP AURACLE). Histologic proof of low-grade glioma was not required for patients with clinical and radiological features consistent with OPG, especially in the case of NF1. In patients for whom a biopsy was performed, the *BRAF* status (V600E mutation, *BRAF*-fusion) was determined if feasible. All eligible patients at the time of treatment had evidence of clinical and/or radiological progression and the following data were collected: reason for treatment, number of prior treatment lines and duration of treatment with bevacizumab. Bevacizumab was administered intravenously (IV) at a dose of 10 mg/kg every 2 weeks. All patients underwent clinical and radiological evaluation: the brain MRI with the closest interval before the first bevacizumab infusion was chosen as the baseline scan. Radiological response was assessed through MRIs performed every three months until the end of treatment. A central radiological review of the images was performed based upon the Response Assessment in Pediatric Neuro-Oncology (RAPNO) criteria for evaluation of pediatric low-grade gliomas [29]: complete response (CR) was defined as complete disappearance of the target lesion, major response as a 75% or greater reduction in three perpendicular planes of the lesion but insufficient to qualify as a complete response; partial response (PR) was defined as a 50% or greater reduction in the target lesion, minor response as a 25–49% reduction in the target lesion; stable disease (SD) was defined as a 0–24% reduction or an increase of less than 25% and progressive disease (PD) as an increase greater than 25% in the target lesion [30]. Radiological control of the disease was defined by the sum of CR + PR + SD. Best responses anytime were considered. Clinical evaluation by trained pediatric ophthalmologists was performed at the same time as the radiological evaluation, with assessment of visual acuity, as well as the visual field (VF) whenever feasible. Teller acuity cards were used in preverbal patients. Standard Snellen chart was used in patients with alphabet knowledge. An equivalent Lea test was used in patients too old to be interested by Teller acuity cards and to young to know the alphabet. Patient vision measurements were initially standardized to decimal measurements and later converted to the LogMAR chart (Logarithm of the Minimum Angle of Resolution) [31]. Taking into account that there is no valid definition of a significant VA change, we based this upon the Ficher et al. definition of a significant VA change and used a 0.2 logMAR change from baseline to define improvement or worsening of the VA [32]. Clinical ophthalmological evaluation was therefore defined as the following: significant VA worsening as a reduction in logMAR scale less than 0.2 units or acquiring blindness; significant improvement as an increase in logMAR scale greater than 0.2 units and steady state if the increase or decrease in the logMAR scale was not sufficient to qualify as worsening or improving. Ophthalmological control of the disease was defined by the sum of CR + PR + SD. A VF evaluation was performed when feasible, but no response criteria were defined. A monitoring of the toxicity of bevacizumab was realized before each administration with clinical and hematological evaluation, blood pressure measurement and a urinary sample checking for proteinuria. The limiting factors for continuation of the treatment were grade 2 proteinuria and grade 2 arterial hypertension according to the Common Terminology Criteria for Adverse Events (CTCAE) v5.0 [33]. Ophthalmologic and radiological response were used to determine the response to the therapy. 

The aim of the work was to evaluate the efficacy of bevacizumab as monotherapy in terms of ophthalmological and radiological response. We expected these results to be comparable to those described with the combination of bevacizumab and Irinotecan as published by Packer et al. [24]. 

### Statistical Analysis

Data analysis was computed with the software Microsoft Excel. Categorical variables were expressed in terms of frequency in percentage with a 95% confidence interval. Continuous variables were expressed as median with minimum and maximum value. For continuous variables, differences between groups were tested with the Student’s t test for normally distributed data.

## 3. Results

Patients’ characteristics and treatment indications. During the study period, 31 patients with OPG were treated with the use of bevacizumab as a single agent. Patients’ characteristics are listed in Table 1.

Fourteen patients (45%) were evaluable for BRAF mutation status; and four tumors (29%) harbored BRAF V600E mutation. Sixteen patients (51%) were evaluable for KIAA1549-BRAF fusion, which was present in the tumors of ten patients (62%). The patient with ganglioglioma was positive for the BRAF V600E mutation. Twenty patients (65%) had not received prior chemotherapy nor radiotherapy. Among patients who had received prior chemotherapy (*n* = 11); three patients (28%) received vincristine and carboplatin according to the SIOP LGG 2004 protocol; four patients (36%) received weekly vinblastine and four patients (36%) received both protocols at a different time.

Indications for treatment, were as follows: VA impairment in twenty-one cases (68%), VF impairment in six cases (19%), strabismus in three patients (10%), Spasmus nutans in one patient (3%). The median duration of treatment was 2.8 months (1.3–11.0), with a median number of six (3–21) bevacizumab infusions.

### 3.1. Imaging Assessment

Radiological response was evaluated in 29 of the 31 patients as MRI images were not available for centralized review in two patients. Almost all patients presented with an optic pathway glioma with chiasmatic localization, and only one patient presented with a left optic nerve glioma. No patient had a radiological CR. Objective response at the 3-month evaluation were the following: one patient (3%) had a 92% reduction (major response); three patients (10%) achieved a partial response; six patients (21%) demonstrated a minor response and stable disease was noted in eighteen (63%). Only one patient (3%) presented with a progressive disease, showing an increase in measurements of 27%. The best anytime responses were as follows: major response for one patient (3%), partial response for six patients (21%), minor response for seven patients (24%), stable disease for fourteen patients (48%) and progressive disease for one patient (3%). The best response was obtained at the first 3-month MRI in twenty-one patients (72%) and at the 6-month MRI in six patients (20%). Both evaluations are summarized in Table 2.

Among the twenty patients (67%) who presented with initial contrast enhancement, fifteen (75%) had a decrease or even a disappearance of the contrast enhancement. Thirteen patients presented with a cystic portion in the target lesion, which decreased with treatment in eleven cases (85%). The patient with a radiological progressive disease showed progression in the cerebral peduncle portion, enhancement of contrast at the 3-month control MRI and the concomitant appearance of a new cystic portion. No ophthalmological evaluation was available for this patient.

### 3.2. Visual Assessment

Visual assessment for visual acuity was available for 25 patients and resumed in Table 3. A graphical version of visual acuity can be found in Figure 1.

Six patients (19%) were not evaluated: three patients were not cooperative enough for VA evaluation due to low age and cognitive status and three patients did not have consistent records of VA. At the start of treatment, ten patients were blind in one eye, eleven patients suffered from optic nerve disk atrophy at fundoscopy and the median LogMAR value before treatment was 0.26 (monocular blindness excluded).

Objective clinical/ophthalmological response to the therapy was the following: steady state in fourteen patients (56%), significant improvement in eight patients (32%), and significant worsening in three patients (12%). Concerning the visual field evaluation, data were available only for twelve patients: seven patients presented with hemianopsia, three patients with quadrantanopia, one patient presented with an unspecified reduced visual field and one patient a normal visual field. Among the three patients who improved on the visual field, two progressed from hemianopsia to quadrantanopia and one improved without more detailed information.

At the follow-up, steady state was achieved in seven patients (59%), improvement in three patients (25%), and worsening in two patients (17%). Only one patient has been treated for amblyopia. Among the eleven patients with optic nerve disc atrophy, nine were stable and two of them aggravated.

### 3.3. Correlation between Radiological and Ophtalmological Response and NF1 Patients

Among the three patients with significant worsening visual acuity, two were with stable disease at the radiological assessment and one presented with a minor response. Among the eight patients with a significative improving sight, four presented with minor radiological responses, one with progressive disease, one with partial response and two with stable diseases. No correlation was found between radiological and ophthalmological responses. Regarding the response to treatment between NF1 patients and non-NF1 patients, there was no difference on either radiological response (*p* value = 0.08) or ophthalmological response (*p* value = 0.06).

### 3.4. Treatment Tolerance and Follow-Up

None of the patients interrupted the treatment for toxicity. One patient presented with grade 3 fatigue/asthenia after three months of treatment and consequently the dose was empirically reduced to 5 mg/kg/dose, with resolution of the symptom. Regarding the follow-up after the bevacizumab treatment: seventeen patients discontinued bevacizumab at the first evaluation (after 3 months), and fourteen patients continued treatment with a maximum duration of 11 months. At the end of treatment with bevacizumab as a single-agent, twenty-two patients received a systematic relay with chemotherapy: twenty-one patients with weekly vinblastine and one with vincristine/carboplatin combination. Patients who finished the treatment with bevacizumab as a single agent did not relapse. Between those who were treated with chemotherapy following the bevacizumab course, nine relapsed and were retreated: seven with bevacizumab alone and two with bevacizumab coupled with vinblastine, obtaining objective responses.

## 4. Discussion

Bevacizumab-based therapies, including bevacizumab plus irinotecan, have been used successfully in children with LGG [24]. In this work, we report a series of 31 patients treated with a bevacizumab single-agent therapy consistent with previous reports of bevacizumab/Irinotecan protocols: we obtained a radiological disease control in 28 out of 29 patients (96%) and an ophthalmological disease control in 22 out of 25 patients (88%).

The primary indication for the use of bevacizumab remains the visual threat and achieving stability in almost all the patients is reassuring. Other radiological findings already described in the literature recur in our series, such as the reduction in the contrast enhancement after treatment and the reduction in the cystic portion in the majority of the patients (85%) [34]. There is only one patient who failed the treatment with real signs of tumor progression and the invasion of the cerebral pendulum by the tumor. Thus, it can be suggested that in this preliminary study there are clear signs of the efficacy of bevacizumab as monotherapy. From a methodological point of view, we based our evaluation of the response to treatment upon a radiological criterion, the reduction in measures of the two and three perpendicular planes of the lesion at the MRI, and an ophthalmological criterion, the LogMAR scale.

In the current literature, in most of the papers, there exists a radiological comparison based on international criteria such those of the RAPNO, and we also based our evaluation on these criteria. Concerning the ophthalmological evaluation, it is difficult to standardize the data. In the literature, there are mainly case reports in which a clinical improvement is described and different scales are used [35]. The most used and appropriated is the logMAR scale [32]. Starting from this assumption, we wanted to evaluate visual acuity in a more objective way and that is why we standardized all the different scales into the LogMAR unit. The use of a 2-line change according to the Ficher et al. definition let us increased sensitivity to early decline in VA [32]. In the case of a stability or amelioration of VA, specificity is lost but the check-in interval of patients (2.8 month) is short and the risk of having an amelioration of VA secondary to other reasons (such as growth) is low. Furthermore, it should be borne in mind that the VA test depends on the patient, and it is subject to a learning curve. In our series, there are only two patients who show a significant improvement in VA and for which, given the young age, we cannot formally exclude a learning phenomenon between the two measurements.

If, on the other hand, we compare the single-agent bevacizumab strategy to the most diffused therapeutic approach combining bevacizumab to irinotecan, we realize right away that results appear similar. Among all, four studies are worth mentioning: the first to describe and report objective radiological response rates as high as 60% were Packer and colleagues, controlling the disease in seven patients out of ten following only two courses of bevacizumab—irinotecan therapy [24]. Through their retrospective study, Hwang and colleagues showed objective responses in twelve out of fourteen patients treated with a bevacizumab based therapy, with rapid clinical benefits, but with a high rate of rebound after treatment came to an end [28]. In their retrospective analysis of fifteen patients with progressive pediatric LGG, Zhukova and colleagues found that all patients with visual deficits had improved or stabilized after a Bevacizumab-based therapy, either coupled with chemotherapy or used alone [21]. Ultimately, De Marcellus et al. obtained disease control using bevacizumab coupled with irinotecan in over 95% of the patients who had previously failed chemotherapy and/or radiotherapy [36]. They also needed to discontinue irinotecan administration in six patients due to digestive toxicity [36]. In fact, among the multiple drug regimens used, bevacizumab is often administered as a third line treatment in combination with irinotecan. Our work suggests that the use of bevacizumab on the front line, with or without a chemotherapy relay, should be evaluated. Furthermore, the treatment leads to rapid clinical improvement and benefits. The bevacizumab anti-angiogenic activity and the resulting decreased blood vessel permeability could play a role in a faster and more effective visual improvement compared to a conventional chemotherapy protocol [21]. The impairment of the VA in these patients is most likely secondary to the direct effect of the tumor size and localization, as well as to a significant inflammatory response/oedema [35]. It is not possible, through this work, to define the duration of the clinical response and the risk of relapse because most of the patients in our cohort had a monotherapy treatment for almost 3 months and, following clinical/radiological response, a chemotherapy relay. Most likely, without any evidence of the absolute necessity to continue the treatment, physicians interrupted bevacizumab immediately after obtaining a response to avoid side effects of the treatment. The decision to carry out an immediate relay with chemotherapy was guided by the objective to maintain a sustained stable disease and prevent rebound, basing this choice on a recent publication [37].

However, it should be noted that three patients benefited from prolonged monotherapy, receiving a median of 20 total doses of bevacizumab (19–21) over a period of 12 months. Although it must be considered that these are only three isolated cases, all three patients demonstrated a recovery of vision and a sustained partial/major radiological response without the need to couple or relay with chemotherapy, pointing to the possible success of a pronged single-agent therapy.

The toxicity profile in our cohort is consistent with previous works and seems acceptable, with mostly reversible proteinuria and hypertension [28,38].

Major limitations of our work are the retrospective model, that does not allow us to have a complete ophthalmological evaluation for all patients and the short average duration of the treatment, which does not allow us to establish PFS curves nor to objectify their effectiveness in a statistical way in the medium to long term. Another limitation is that most of our patients were treated for visual deterioration (both visual acuity and visual field) and as the European report already demonstrated, patients with recent visual deterioration prior to treatment responded better than those with established poor vision but no evidence of recent deterioration [18].

## 5. Conclusions

In conclusion, our study allows us to confirm the effectiveness of bevacizumab as monotherapy, both from a radiological and clinical/ophthalmological point of view.

In our cohort, there was little correlation with visual outcome and radiological evaluation. Future trials of bevacizumab could reasonably be monitored with visual outcomes alone, as this is the main concern of parents of children with optic tract glioma.

Since most of these patients are chronically ill, with a high burden of morbidity, it is essential to find a sustained, prolonged and adequate therapy without major toxicities, which could be conducted by removing chemotherapy and then avoiding irinotecan-induced diarrhea, and administering bevacizumab as a single agent. Prospective multicentric data of a bevacizumab monotherapy with a detailed ophthalmological pre-treatment status of the patients would, however, be required in order to consider its functional benefits.

## Figures and Tables

**Figure 1 cancers-15-01036-f001:**
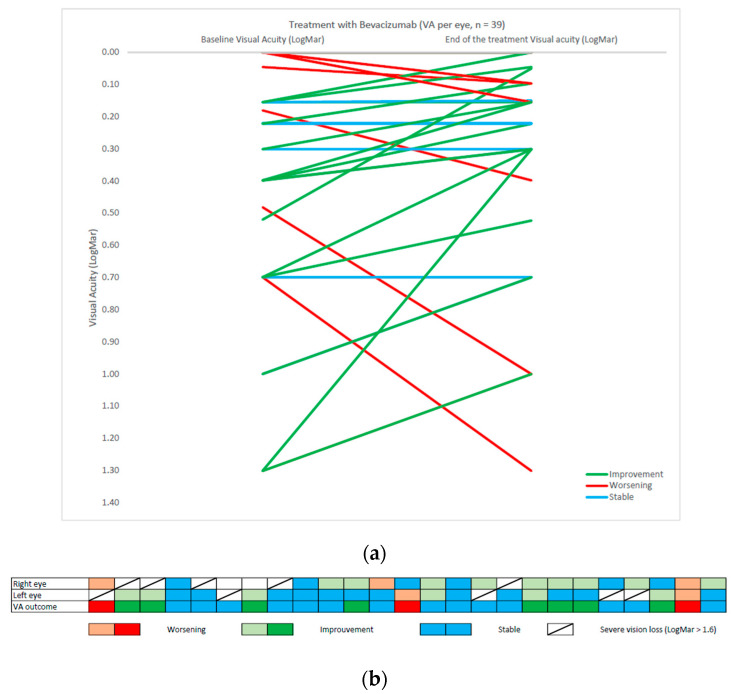
(**a**) VA graphical outcome per eye for 25 patients in baseline and after treatment with Bevacizumab: 39 evaluable eyes, monocular blindness excluded. (**b**) VA outcome represented with a column for each patient. Right and left eye evaluation and binocular outcome according to the graphical representation standardized by the SIOPE NF1 OPG workshop.

**Table 1 cancers-15-01036-t001:** Clinical characteristics in 31 patients treated with bevacizumab monotherapy.

Clinical Characteristics	No. of Patients
Total number of patients	31
Median age at diagnosis (range)	3.8 years (0.3–19.0 y)
Median age at first bevacizumab dose (range)	4.9 years (0.3–19.0 y)
No. of patients with NF1	11 (35%)
Histology	17 patients (57%)
Pilocytic astrocytoma	16 (52%)
Ganglioglioma	1 (3%)
Prior chemotherapy	11/31 (35%)
Weekly vinblastine	4 (36%)
SIOP LGG 2004(vincristine—carboplatine)	3 (28%)
Both chemotherapy protocols	4 (36%)

**Table 2 cancers-15-01036-t002:** Objective radiological responses according to RAPNO criteria at 3-month MRI and best anytime response during the treatment period, expressed as number of patients and percentage.

Radiological Response	3-Month Evaluation	Best Anytime Response
Complete response	0 (0%)	0 (0%)
Major response	1 (3%)	1 (3%)
Partial response	3 (10%)	6 (21%)
Stable disease	18 (62%)	14 (48%)
Minor response	6 (21%)	7 (24%)
Progressive disease	1 (3%)	1 (3%)

**Table 3 cancers-15-01036-t003:** Baseline ophthalmologic data and change in function after treatment.

Clinical Characteristics at Start of Treatment	No
Visual acuity evaluable (*n* = patients)	25
Monolateral blindness (*n* = eye)	10
Median (LogMar)	0.26 (−0.08–1.30)
Optic disk atrophy at fundoscopy (*n* = patients)	11
Clinical characteristics after bevacizumab treatment	No
visual acuity evaluable (*n* = patients)	25
Monolateral blindness (*n* = eye)	9
Median (LogMar) at the end of treatment	0.22 (−0.08–1.30)
Median (LogMar) at best moment	0.15 (−0.08–1.30)
Optic disk atrophy at fundoscopy (*n* = patients)	11
Change in Visual Acuity (VA)	No of patients (Percentage)
Significant improvement (>0.2 LogMAR)	8 (32%)
Steady state (change within 0.2 LogMAR)	14 (56%)
Significant worsening (<0.2 LogMAR)	3 (14%)
Change in Visual Field (VF)	No of patients (Percentage)
Improvement	3 (25%)
Steady state	7 (59%)
Worsening	2 (17%)

## Data Availability

The data that support the findings of this study are available from the corresponding author, upon reasonable request.

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
