# Peer review of "Bevacizumab as Single Agent in Children and Teenagers with Optic Pathway Glioma"

_cancers, 2023, doi:10.3390/cancers15041036_

Round 1
Reviewer 1 Report
This is a retrospective cohort study evaluating the impact of single agent Bevacizumab upon radiological criteria and visual outcome criteria in children with optic pathway glioma. Despite a ten year study period there are only 31 patients for analysis.
The cohort is not well described as the only age data is the age at initial presentation. It is not clear whether this is the age at treatment with bevacizumab as they refer to cases where multiple previous treatments had been used.
Bevacizumab has previously been used in conjunction with Irinotecan for the treatment of optic pathway glioma. Irinotecan is a cytotoxic agent with significant bone marrow and GI side effects. The authors are interested to report on their experience of using this anti-angiogenic drug as a single agent.
They have not specified the anticipated mechanism(s) of action of Bevacizumab in either reducing tumour size or saving vision. This is important as vision preservation or improvement is dependant upon neuronal function. A mechanism by which VEGF blockade preserves neuronal function would be an important consideration in the design of this study and interpretation of results.
The authors have not cited the following European OPG Workshop report in their introduction or considered it in their discussion. It is directly relevant to their study and offers a doubling in evidence on visual outcome when added to Fisher's report from the USA:
NF1 optic pathway glioma. Analysing risk factors for visual deterioration & indications to treat Amedeo A. Azizi, MD 1, David A. Walker, BMBS 2, Jo-Fen Liu, MSc 2, Astrid Sehested, MD 3, Timothy Jaspan, MB ChB 4, Ian Simmons, MB ChB 5, Rosalie Ferner, MD 6, Jacques Grill, MD PhD7, Darren Hargrave, MD 8, Pablo Hernáiz Driever, MD9, D. Gareth Evans, MD 10 and Enrico Opocher, MD 11 on behalf of the SIOPE NF1 OPG Nottingham, UK, Workshop 2014 12 Neuro-Oncology, https://doi.org/10.1093/neuonc/noaa153 July 2020
(File attached)
The cohort that has been assembled by the authors is representative of centre based cohort. However not all cases were evaluable with both radiological and visual assessments. There is insufficient information about the pre-treatment status of each case with respect to visual symptomatology such as optic atrophy, visual fields, visual acuity measurement techniques across age groups or Logmar scores data. Logmar data is referred to, but is not shown. There is no reference to OCT assessments which are currently recommended for assessment of such cases as a recognised measurement of neuronal functioning. Similarly the imaging data is not complete as there is no data on the modified Dodge classification, tumour dimensions or the way they were measured.
The data is not presented in graphical format. The European Workshop Report identified standardised proposed methods for reporting, ideally these should be replicated. There was no attempt to correlate imaging assessment with visual response, which has previously been shown to correlate poorly in both Fisher's and the European report.
This means that the conclusions of the European Workshop Report which was based upon a multi-centre international cohort of patients from the SIOP LGG 2004 trial, regarding selection of patients for drug therapy were not included in this cohort study's design or analysis.
Drug treatment for OPG is used primarily to save vision. The European report demonstrated that those with recent visual deterioration prior to treatment responded better than those with established poor vision but no evidence of recent deterioration.
It would be important that this and future studies measuring visual outcomes report the pre-treatment status of the patients in detail in order to ensure that apparent response rates are compared for similar pre-treatment indications. As written this study fails to meet that requirement and is not suitable for publication without significant modification.

Reviewer 2 Report
This is retrospective study conducted in children and adolescents with OPG treated with bevacizumab as a single agent. The study aims to demonstrate the effectiveness of bevacizumab as monotherapy, both from a radiological and clinical point of view.
Although it is an interesting study focused on a very rare tumor, the retrospective nature of the study, the small sample of patients and the short average duration of the treatment make this work inadequate to provide reliable PFS data; however, it provides the suggestion of the potential efficacy of bevacizumab in this population at cost of an acceptable toxicity profile.
Although the results of the study are not robust from a statistical point of view, I consider the study interesting and publishable considering the rarity of this disease.
Round 2
Reviewer 1 Report
The authors have responded to most comments. The literature basis of their work is improved. They have developed a graphical representation of the data. They have acknowledged the limitations of their data. A remaining criticism is that this is a third study demonstrating that imaging can identify changes in tumour dimensions with poor correlation with visual outcome. One conclusion could be that future trials of bevacizumab could reasonably be monitored with vision outcomes alone as this is the primary concern of parents of children with optic pathway glioma. They do not indicate the reasons for early discontinuation of bevacizumab or record toxicities this should be added
The figure 1 needs reconsideration. The report by Azzizi suggested standardised approach to reporting using LogMar grids for left and right eyes before and after treatment. This is the new standard and should be adopted.
I was unable to interpret Fig 1a and the coloured lines in Fig 1b were not identified in the legend
Azzizi's paper used the before and after lines to compare those with recent or established vision loss. The lines format could be best applied to study the impact of pre treatment vision changes.
